# What Is the Role of Night-Time Noise Exposure in Childhood Allergic Disease?

**DOI:** 10.3390/ijerph19052748

**Published:** 2022-02-26

**Authors:** A-Ram Kim, Jin-Hee Bang, Sung-Hee Lee, Jiho Lee

**Affiliations:** 1Department of Occupational and Environmental Medicine, Ulsan University Hospital, University of Ulsan College of Medicine, Ulsan 44033, Korea; 0735454@uuh.ulsan.kr (A.-R.K.); shee5173@gmail.com (S.-H.L.); 2Environmental Health Center, University of Ulsan College of Medicine, Ulsan 44033, Korea; jhbang0326@naver.com

**Keywords:** noise, allergic disease, sleep disturbance

## Abstract

The cause of the allergic disease is known to be multifactorial, and there is growing evidence of environmental factors triggering the disease. Indeed, it is essential to find modifiable environmental factors related to allergic disease. Noise is an environmental pollutant causing various health problems, especially when exposed during the night-time. This study assessed the impact of night-time noise exposure in allergic disease. Subjects were selected from a panel data survey containing questions on allergic disease and related factors. Incidence of allergic disease, covariates, and addresses was derived from survey questionnaires. By applying the Land Use Regression modeling method, each subject’s night-time noise estimates were elicited based on the night-time noise level collected from the noise monitoring site. Association between night-time noise difference rate and incidence of asthma were analyzed by Cox proportional hazard regression. Incidence of allergic disease increased when night-time noise difference was positive compared to the negative difference. Additionally, the incidence of allergic disease increased by per interquartile range of night-time noise difference rate. The result showed that exposure to night-time noise tends to increase the risk of allergic disease. With further studies, the result of our study may serve as supplementary data when determining noise limits.

## 1. Introduction

Allergic disease refers to a group of diseases including asthma, allergic rhinitis, and atopic dermatitis, those regulated by an immune mechanism involving inflammatory cells, cytokines, and neuropeptides [1,2]. As these group of diseases shares common pathophysiology, it is referred to as “atopic march”, implying a particular disease at risk at a specific time frame of childhood [3]. The two important risk factors in the development of allergic disease are genetic factors and environmental factors. The most well-known environmental trigger factors of allergic disease are irritants, aeroallergens, food, microbial organisms, sex hormones, stress factors, sweating, and climatologic factors [1,4].

Since 1970, the prevalence of allergic disease has been in an increasing trend worldwide. In South Korea, the ten-year trend (2008–2017) in the incidence of allergic disease differs by specific disease [5]. The incidence of allergic rhinitis increased regardless of the age group, but in the case of atopic dermatitis, the occurrence increased in the elderly group, when in infants and preschool children it decreased [5]. On the other hand, asthma showed a decreasing pattern in both adults and children [5]. Allergic disease is continuously drawing attention because the nature of symptoms impairs the quality of life and result in high medical costs in individuals with allergic disease [6,7,8]. As avoiding aggravating factors is one of the fundamental principles in managing allergic disease [9], avoiding triggering factors could be a crucial measure in preventing allergic disease. Indeed, to establish control or preventive measures, identifying the predisposing factor and aggravating factor of allergic disease is crucial for managing the disease. The main pathophysiology of allergic disease can be described as immune dysregulation, which leads to harmful chronic inflammation [10]. Recently, there have been reports that interaction between nervous, endocrine, and immune systems plays a pivotal role in the development of the disease. Thus, the role of psychological stress has been emphasized both in the development of the disease and aggravation of its symptom [6]. Ising et al. reported that adjuvant factors such as air pollutants and noise could stimulate the occurrence of allergies [11].

Noise, defined as unwanted sound, is a growing issue for many countries, causing noise pollution [12]. The main source of noise affecting people is traffic noise [13]. Noise can affect health in many aspects, including auditory and non-auditory health effects [11,14]. Previous studies on noise and health outcomes reported that noise exposure could cause hearing impairment, annoyance, cardiovascular disease, lower job performance supported by sufficient scientific evidence [15,16]. WHO Europe reported that sleep disturbance is the major health outcome of environmental noise exposure followed by annoyance with regard to the “burden of disease” [17]. According to Dang-vu et al., human brain takes in environmental sound, exerting physiological responses on motor and autonomic systems and even cortical arousal [18]. Additionally, previous studies suggest that exposure to noise at night-time is more relevant with the long-term health outcome than exposure to noise at daytime [19]. Furthermore, the impact of noise on health is thought to be exceptionally debilitating on vulnerable people, including children and the elders [20]. *Noise impacts on health* (*Science for Environment Policy*) stated that when children younger than ten years old were exposed to noise, hospital admissions for respiratory disease including pneumonia and asthma attacks in asthmatics (girls only), noise annoyance, sleep disturbance, heart and circulation problems, lower perception of quality of life due to the increased stress hormone, lower job performance, and hearing loss and tinnitus increased [20]. These health outcomes are estimated to be distinct when subjects were exposed to night-time noise [20].

Difficulties in assessing noise exposure at the individual level made it difficult to conduct a large-scale study of noise exposure on health [21]. Several methods (interpolation, statistical model, air quality model) were applied to predict the noise levels at unmeasured sites [22,23,24,25]. A statistical model can improve the accuracy of the noise exposure assessment for a population in a complex urban environment [26,27,28]. Land use regression (LUR) modeling is currently one of the most used methods for assessing exposure to air pollution in epidemiological studies [29,30,31]. In this study, we first predicted noise levels at locations of subjects using the land use regression (LUR) model based on noise levels provided from monitoring sites. With these night-time noise estimates at the individual level, we analyzed the impact of exposure to night-noise estimates on the incidence of allergic disease.

## 2. Materials and Methods

### 2.1. Sububject Selection

The Environment Health Care Center of Ulsan University Hospital has established an elementary student cohort from 4 elementary schools with different background environments (residential area, industrial area, coastal area, downtown area), and has been conducting survey questionnaires based on the International Society of Asthma and Allergy of Children (ISAAC) since 2009. Students from 4 elementary schools participate in the survey, two schools at 2-year intervals. The location of each participating elementary school is presented in Figure 1. The background environment represented by each school is as follows: Gulhwa represents the coastal environment, Samshin represents the downtown area, Myeonchon represents the residential area, and Yangi represents the industrial area. A trained researcher who belongs to the Environment Health Care Center of Ulsan University Hospital reviewed surveys and formed the computerized dataset. For each round of the survey, survey questionnaires were double-checked to look for any invalid responses to the question.

We assessed data from June 2009 to April 2018, which resulted in a total of 5 datasets for a 9-year period. Parents received the questionnaire through the school and completed the given questions. They were encouraged to send back the completed questionnaire within one week to apply the same air pollutant dataset to the subject group who conducted the survey at the same period.

Inclusion criteria were: (1) participate in more than one survey; (2) complete the question “Have you been diagnosed with any of the following diseases last year?; atopic dermatitis, asthma, or allergic rhinitis”. Specific exclusion criteria were as follows: (1) underlying allergic disease determined by the question “Have you ever diagnosed with any of the following diseases once?; atopic dermatitis, asthma, or allergic rhinitis” at their first survey; (2) missing answers on questions asking “address”, “sex”, and “age”. Based on these criteria, a total of 2972 students remained. Among those, 670 students were diagnosed with allergic disease during the study period, while 2302 students were not (Figure 2).

All parents of the study subject were provided written informed consent prior to participation with sufficient explanation of the survey. This survey was approved by the Institutional Review Board of the Ulsan University Hospital (approval no. UUH 2009-09-061-022).

### 2.2. Exposure Assessment

#### 2.2.1. Night-Time Noise

We performed a model each year (2009–2018) to predict the annual average night-time noise level. The LUR model utilizes the monitored concentration as a dependent variable. We used National Noise Information System data [32]. Night-time monitoring is conducted twice a day (11 p.m., 1 a.m.), and quarterly data is released. The LUR models employed geographic variables of road density, land use, distance to the nearest main road and highway, and altitude as noise monitoring sites (Figure 3). Land use data were used Environmental Geographic Information Service [33]. Elevation data were used Shuttle Radar Topography Mission 3s (SRTM) data [34]. Geographic data were used to calculate the total length (km) of road and area (km^2^) in buffers of 50, 100, 300, 500, 1000 m, the distance to the nearest main road and highway.

The LUR model is based on the principle that spatial distribution of noise levels is correlated with the local environmental variables, and the relationship between noise levels and environmental variables is predicted through multiple regression analysis [35,36,37,38]. In general, a multiple linear regression model can be expressed as: Y_i_ = β_0_ + β_1_X_1_ + β_2_X_2_ + β_3_X_3_ + … + β_n_X_n_ (Y_i_: noise level, β_0_: constants, β_n_: respective coefficients, X_n_: covariables dependent variables). The LUR model is used to predict the concentration of unmeasured locations based on the predictor variables. The results of the LUR model are the annual average night-time noise levels at the locations of the questionnaire participants. ArcGIS pro tool Version2.7.0, esri Korea was used to generate independent variables. The models were constructed using IBM SPSS statistics for Windows Version 20.0. (IBM, Armonk, NY, USA).

The LUR model and spatial interpolation method were compared to evaluate the spatial distribution of estimated noise levels. As kriging is one of the most frequently used interpolation methods [39,40,41], kriging was used in this study.

#### 2.2.2. Air Pollution Level

The values for estimating air pollution concentrations were based on the levels produced by Kim et al. (2021)’s study [42]. Air pollution estimates were based on the monthly average data from 13 air quality monitoring sites (AQMS) under the National Institute of Environmental Research (NIER). From the measured data, air pollution estimates for each school level were drawn by using the Community Multiscale Quality (CMAQ) model (version 5.0.1) predictions. The CMAQ model setup was based on the CMAQ model setting used in Kim et al. (2021)’s study. Brief descriptions of the CMAQ model used are as follows: time of the model was mid-year in 2014, location encompassed Ulsan with target region at the fine-scale innermost domain and rest being surrounding area. Moreover, CMAQ-ready meteorological and emission inputs, initial and boundary conditions, and physical and chemical options were used [35]. After calculating 1 km gridded concentrations predicted by the CMAQ model each hour averaged on a monthly basis, it was blended with ambient monitoring data from 13 air quality monitoring sites in Ulsan (AQMS) (Figure 2), which involves a combination of the inverse distance weighting (IDW) method for spatial monitored data interpolation and spatial scaling using gridded CMAQ predictions [35,36]. As the concentration data provided by the monitoring site could differ considerably from the actual concentration of the school area, adjusted concentrations can be derived by fusing monitoring data with CMAQ prediction, which serves as a more reliable concentration of four school regions. Thus, these fused gridded concentrations of school regions could be used as concentration estimates that subjects from each school are exposed to. This fusion technique was used in various past studies to improve the CMAQ predictions for air pollution exposure estimation [35,36,39,40]. The result of the air pollutant concentration is presented in Figure 4 for each round of the survey and the schools.

### 2.3. Definition of the Outcome

In this study, the incidence of the allergic disease was defined as the self-reported of one or more doctor-diagnosed atopic dermatitis, asthma, or allergic rhinitis one year before the participation. Incidence of allergic disease was recorded as the event and was based on the question “Have you been diagnosed with any of the following diseases last year?; atopic dermatitis, asthma, or allergic rhinitis”. Answering “yes” on that question was defined as the occurrence of the event.

The covariates considered on the analysis were the general factors known to be associated with allergic disease from past studies, which was categorized into individual-level demographic (family income), or intrinsic factors (sex, age, subjects’ past medical history, and family history of allergic disease) and environmental exposures. Environmental factors such as air pollutants (NO_2_, SO_2_, O_3_, CO, PM_10_) were obtained from the air pollution monitoring site and modeled to each subjects’ school area. History of ever owning a pet, use of air purifiers and humidifiers, exposure to secondhand smoking were also included as the indoor source of environmental exposure and amount of traffic near the residence; distance to the road was included as the outdoor source of environmental exposure.

### 2.4. Definition of the Exposure

Variables to determine air pollution exposure were defined as differences of each air pollutant level between the year of the last survey and that of the initial survey. Two distinct variables were created to perform 2-step regression model to test the impact of night-time noise exposure on health. The first variable included in the first model was the difference of the estimates between the last survey and first surveys, the negative difference being the reference value and the positive difference being the compared value. The second variable included in the second model was the difference rate divided into interquartile ranges serving the lowest interquartile range, it being a reference to examine whether the change in the rate of the pollutant exposure level affects health.

### 2.5. Statistical Analysis

Cox regression analysis with a backward stepwise likelihood ratio entry method was performed to analyze the association between exposure to night-time noise and incidence of allergic disease, with censoring at the time of the last survey of each subject. Air pollution and night-time noise exposure variables were included as the difference of the estimates between the last survey and the first survey. Using these difference variables in the first model, we could identify that the difference in night-time noise exposure was positively associated with the incidence of asthma. In the second model, we included the night-time noise variable as the difference rate. In the univariate model, only a single variable was included as an explanatory variable. The multivariate model adjusted for various covariates, including age at entry, sex, income, air purifier, parental history of allergic disease, history of bronchiolitis within two years of birth, history of oxygen therapy at birth, preterm birth, ever having experienced a daycare center, ever having moved to a new house, exposure to second-hand smoking, NO_2_ difference, SO_2_ difference, CO difference, O_3_ difference, PM_10_ difference, ever having had a pet, amount of traffic, distance to road, and use of humidifier [10,14,43,44,45,46,47,48,49,50,51,52]. The covariates adjusted for in the second model were the same besides the variables on night-time noise, NO_2_, SO_2_, CO, O_3_, and PM_10_, as night-noise difference rate in interquartile range, NO_2_ difference rate, SO_2_ difference rate, CO difference rate, O_3_ difference rate, and PM_10_ difference rate. The results are presented as hazard ratio (HR) and 95% confidence interval (CI) for per interquartile range (IQR) increase for night-time noise difference rate.

IBM SPSS statistics for Windows Version 20.0. (IBM, Armonk, NY, USA) and STATA SE Version12, (StataCorp LLC, JasonTG, Seoul, Republic of Korea) were used, and a *p*-value less than 0.05 was interpreted to be a statistically significant result.

## 3. Results

The mean follow-up period in subjects who developed allergic disease was 865.2 days (ranging from 2 to 3 rounds). During the follow-up period, 29.1% (670) had been diagnosed with allergic disease. The mean age at their first survey was 7.5 ± 1.77 years (Table 1). Subjects who developed allergic disease tend to be younger, have a parental history of allergic disease and past history of bronchiolitis than subjects without allergic disease. There was no significant difference in family income between the two groups. Both indoor and outdoor environmental exposure factors between the two groups did not show statistically significant differences except for the variable “possessing air purifier at home” (Table 2).

### 3.1. Air Pollutant Estimates

Air pollutant estimates were based on the location of each school and study year. The temporal trend of air pollutants by each school is illustrated in Figure 4. The pattern of the air pollutant level does not show a distinct temporal trend nor variability across schools (Figure 4).

### 3.2. Night-Time Noise Estimates

In this study, the LUR model was constructed using the noise monitoring data and geographic information in Ulsan. Variables with low correlation were rearranged through stepwise linear regression analysis in estimating the noise level at the location of the subjects in the questionnaire. The annual average noise level from 2009 to 2018 at each subjects’ location predicted through the LUR model was in the range of 51.4 to 56.2 dB (Figure 5). The model R^2^ explaining the variability in the noise levels for all LUR models ranged from 0.48 to 0.59. The LUR models for noise levels largely included commercial, industrial variables with 1000 m buffer, road length with 50 m, and 1000 m buffer. Comparing the time trend in noise level by each school, the overall annual noise level by subjects from each school decreased over the time period (Figure A1), although there was a slight difference by the school (Figure A2). We found out that most levels exceeded the permissible noise limits of 40 dB of the general residential area, and some of them even exceeded the permissible noise limit of 55 dB of the roadside area under the Framework Act on Environmental Policy.

When compared with the concentration from kriging (Figure 6), the concentration derived by the LUR model more accurately reflected the land use, showing higher concentrations at the intensive traffic network (highway and main road) and the commercial/residential areas. In kriging results, high noise levels were simulated in an area without a noise source (Figure 6, river), whereas the LUR model predicted improved results.

### 3.3. Correlation between Night-Time Noise Estimates and Air Pollitants

The correlations between night-time noise and air pollutants were observed with a Pearson correlation coefficient ranging −0.037–0.510 (Table 3). Except for the correlation between NO_2_ and SO_2_, all other variables showed weak or negligible correlation. The correlation between NO_2_ and SO_2_ showed a moderate negative correlation. The night-time noise estimates showed decreasing pattern by time in all four schools. Under the Framework Act on Environmental Policy of South Korea, the permissible night-time noise is set at 40 dB at the general residential area and 55 dB at the roadside. Furthermore, under the Noise and vibration control act of Enforcement rule, article 25 (Road traffic noise and vibration management guideline), the permissible night-time noise is 58 Leq dB (A). The estimated levels of most study subjects exceeded 40 dB, and some even over 55 dB. The percentages of exceeding the night-time noise limit of the general residential area by the school are as follows: Gulhwa 99.9%, Myeonchon 98.1%, Yangji 100%, and Samshin 100%. The percentages of exceeding the night-time noise limit of roadside by the school are as follows: Gulhwa 3.9%, Myeonchon 5.3%, Yangji 29.2%, and Samshin 29.3%.

### 3.4. Association between Night-Time Noise Exposure and Incidence of Allergic Disease

The group with a positive difference of noise level significantly increased the risk of allergic disease by 7% based on a hazard ratio of 1.077, compared to the negative value of noise difference (95% CI 1.057–1.099). Even after adjusting other covariates, the risk of allergic disease incidence increased significantly by 71% (adjusted HR 1.710; 95% CI 1.424–2.052, *p* < 0.05) (Table 4). From the result of the first model, it can be interpreted that subjects who experienced an increase in noise level might be prone to the development of allergic disease. So, for further analysis, the difference rate of night-time noise level was categorized into interquartile range, and cox regression was performed as the second model. Subjects with difference rates in quartiles 3 and 4 showed significantly higher risk for incidence of allergic disease compared with quartile 1 both before and after adjustment (Table 5). However, the strength of association decreased slightly after the adjustment.

Other variables that increased the risk of allergic disease are as follows: age, history of father’s allergic disease, mother’s allergic disease, past history of bronchiolitis, possessing air purifier, and higher NO_2_ difference rate. In terms of income, the risk of allergic disease significantly decreased with higher income categories compared to the lowest income category.

## 4. Discussion

Compared to subjects who experienced lower night-time noise levels in subsequent study years, those who experienced higher levels had a higher risk of being diagnosed with allergic disease. This finding remained significant after adjusting factors known to be associated with the incidence of atopic dermatitis, including individual-level factors, indoor and outdoor environmental factors of residence such as air pollutants (NO_2_, SO_2_, O_3_, CO, PM_10_), and secondhand smoking. From the result of the first model, it could be assumed that a night-time noise increase can make related subjects vulnerable to the development of allergic disease. For the next step, in the second model, we analyzed to look into how the extent of the noise difference rate affects allergic disease incidence. It turned out that allergic disease risk increased with a higher noise difference rate demonstrating a dose–response relationship.

Noise, as mentioned previously, cause not only hearing impairment but also affects multiple organ systems of human [19,47,53]. Many epidemiological studies have suggested the nonauditory effect of long-term noise low-level noise exposure, including cardiovascular disease, respiratory disease, and psychological disease [13,19,53,54,55,56,57,58,59,60]. Sleep disturbance and mental health problems are among the seven health effects and social outcomes from noise exposure confirmed by WHO [56]. Especially night-time exposure to noise is considered more serious because it directly influences sleep architecture along with sleep quality resulting in a detrimental effect on the quality of life [20,61]. Accordingly, sleep disturbance is one possible mechanism of our result, noise impact on allergic disease.

It is well known that the hypothalamic-pituitary-adrenal (HPA) axis and cortisol play a significant role in regulating the stress response, and sleep has a potent inhibitory influence on this pathway. Thus, sleep disruption influences health by interrupting the HPA axis and autonomic sympathoadrenal system, resulting in numerous health outcomes [62,63,64]. Typically, hormones secreted by the HPA axis follow a circadian rhythm. In the case of cortisol, the lowest concentration occurs during the first phase of sleep, which is about midnight, and slowly arises, reaching peak level in the early morning [65]. Thus, night-time noise exposure, especially during the first phase of sleep, the normal nadir for cortisol and the highest growth hormone level is interrupted, disrupting the circadian rhythm, leading to poor physical and psychological recovery [65]. Halperin et al. reported that even lower noise at night could trigger physiologic reactions, including increased hormone secretion, body movement, and cortical arousal, which lead to sleep fragmentation [56]. As the HPA axis functions in a bidirectional fashion with sleep, arousal leads to dysfunction of the HPA axis in turn, activating the secretion of the so-called stress hormone causing health impairment which can be regarded as an indicator of chronic stress [61,62,66,67,68,69]. In addition, noise, as a nonspecific stressor, activates the autonomic nervous system directly and elicits subsequent endocrine signaling by disrupting the sleep process, eventually evoking stress response which activates the autonomic and endocrine system and subsequently generates chronic stress [59]. Supporting this result, a cohort study of children from Stockholm that traffic noise exposure was associated with the saliva cortisol levels [19].

Another mechanism suggested by Prasher is that noise is a stressor itself. Noise alters the self-regulating process of our biological system to keep the stability of one’s body through disruption of the endocrine and immune system [70,71,72]. Previous studies have reported that noise-induced stress exposed to both acute and chronic noise increases the cortisol level or disrupts the regulatory mechanism of the cortisol by affecting the HPA function. In the past, some form of allergic disease was expressed by terms related to psychologic origin [65]. For example, atopic dermatitis was once referred to as dermatitis nervosa, indicating psychological factors have a role in the pathogenesis of the allergic disease. According a study by Kitagaki et al., psychological stress was related to the development of atopic dermatitis [73]. Other studies found that atopic dermatitis, one of the allergic diseases, can be triggered and aggravated by both physiological and psychological stress [74]. Noise induces both physiological stress and psychological stress [75]. Sleep disturbance has a complex relationship with immune system cortisol rhythm playing the primary role [76,77]. Cortisol is known to be a potent immune and inflammatory suppressor, which is supported by the secretion of various inflammatory cytokines [64]. Furthermore, sleep disturbance increases oxidative stress, leads to a pro-inflammatory state by increasing secretion of IL-1 and TNF-a, and activates the sympathetic nervous system [78,79]. Additionally, sleep disturbance and stress could shift the TH1/TH2 balance by disturbing the functional rhythm of regulatory T cells [78]. These overall mechanisms of noise effects confirm that stress-induced hormonal secretion is a possible proxy of noise-related allergic disease. Additionally, epidemiological studies have found noise-induced sleep disturbance was related to psychiatric symptoms, behavioral problems, allergic disease, asthma, skin disease [80,81]. A longitudinal population-based study confirmed that children who are overtired showed an increased risk of rhinitis symptoms five years later with an adjusted odds ratio of 2.59 (95% CI; 1.31–5.11) [82]. Comprehensively, night-time noise is associated with the development of allergic disease by complex interaction with sleep disturbance, neuroendocrine system, immune system, and autonomic system [6].

In our study, the noise estimates of the subjects were mostly over 40 dB; some of them even exceeded 50 dB. From this result, it can be assumed that subjects of our study had suffered from sleep disturbance regardless of whether they were aware of the symptom or not, responsible for the development of allergic disease to some extent [20,83]. What is evident from our result is that when a subject experiences louder noise compared to the baseline, they are more likely to develop allergic disease.

Other than the night-time noise level, the result of air pollutants was not consistent with past studies. A panel study analyzing the short-term effect of air pollution on atopic dermatitis symptoms in children reported that an increase in PM_10_, NO_2_, and O_3_ was positively associated with increased atopic dermatitis symptoms, which is inconsistent with our result, except for the NO_2_ [9]. The different processes of estimating the air pollutant concentration could have resulted in this inconsistency between the results. It was estimated at the school level rather than estimating each air pollutant concentration at each subject’s level. Additionally, the long-term effect of environmental pollution is relatively more challenging than investigating the short-term effect on health. Because above all, exposure assessment is complex. In addition, it is known that the effect of long-term low levels of environmental exposure on health varies significantly among people due to other covariates such as genetic factors. Supporting this fact, the parental history of the allergic disease strongly increased the risk of allergic disease in our study, consistent with previous studies. One interesting result was that possessing an air purifier increased the risk of allergic disease. This result may reflect the cross-sectional nature of this study, meaning the chronological order of the event cannot be determined from the survey data. Nevertheless, the result can be explained that parents of children who developed allergic diseases might have bought an air purifier with the hope to alleviate the symptom.

WHO recommended maximal night-time noise in a bedroom of 45 dB and a mean level of 30 dB [13], but even with noise under those limits, health effects can occur through the mechanism described above. Noise, recognized as an air pollutant, is a growing concern for causing harm to millions of peoples’ health and lowering the quality of life among many countries, including Europe, the United States (US), and the Republic of Korea [84]. The European Environment Agency reported that more than a 100 million people are affected by the harmful level of noise, road traffic noise being the most widespread source [85]. In the US, the Environmental Protection Agency regulates the noise source under the Clean Air Act (Title IV—Noise pollution) [86], meaning that noise is considered an air pollutant and should be regulated and controlled as other air pollutants. Due to ongoing industrialization and urbanization, transportation means have become a crucial part of human life, and the United Nations (UN) predicted that 68% of the human population wants to live in an urban environment [87]. The report released from “Korea Environmental Institute” in 2019 shows that due to the increase in the number of vehicle registrations and length of road each year, traffic load and flow are rising trends making traffic noise inevitable [88]. In addition, as the development and supply of housing complexes in the area around the highways increases, conflicts from roadside traffic noise continue to occur [88]. From the reports of the 2001 census of road traffic noise exposure in Korea, 52.7% of the population exposed to noise above 55 dB(A) at night-time (22:00~06:00) was found to have exceeded the noise environment standard [88]. Thus, noise issues are expected to grow continuously due to traffic and neighborhood noise. Considering that children are more vulnerable to night-time noise exposure, there is a need to manage traffic noise at night.

This study has several limitations. The major limitation is related to the estimation of the night-time noise level. First, the night-time noise level was estimated based on the monitoring data. As the noise monitoring sites are located outdoors, it reflects the noise coming from the outdoor environment, mainly the industrial and traffic source. In the Republic of Korea, neighborhood noise accounts for most of the complaints from environmental noise and vibration. Thus, estimating the noise exposure at the individual level would be more appropriate. However, on the other side, statistics on “Cause of Environmental Disputes Application from 2008 to 2018” revealed that dispute cases on environmental noise and vibration from industrial and traffic sources account for more than 50% of total cases. Although most of the dispute cases are from an industrial source, cases of traffic noise have increased 16.9% in 2017 compared to that of 2016, drawing attention to the necessity for managing traffic noise [88,89]. In addition, there were only 87 noise monitoring sites, and the level of the height where the subject is living was not considered in LUR modeling. For example, if subjects ‘a’ and ‘b’ live in an apartment near ‘noise monitoring site X’, they would get the same night-time noise estimates even if they live on a different floor. In brief, the LUR model can only calculate the extremely local variations in noise level. Nevertheless, as shown earlier in Figure 6, night-time noise estimates derived by the LUR model simulate the surroundings more reasonably than the simple interpolation method. Second, as the analysis is based upon the self-reported questionnaire, there were missing values, and the subjectiveness of the answer cannot be ignored. However, this limitation was compensated for by using the question item asking for the history of the doctor’s diagnosis last year. Third, meteorology was not considered as a covariate. As sound waves travel through the atmosphere, refraction and diffraction of the sound waves are affected mainly by meteorologic factors such as wind speed, temperature, wind direction, and turbulence [90]. That is why individuals who live close to highways experience a different level of traffic noise from time to time. In addition to its effect on noise propagation, previous studies have reported that humidity and temperature can trigger allergic disease [9]. So, in the future study, meteorologic information should be considered. Over and above considering meteorologic information in the future study, other environmental factors that could have an impact on the development of the allergic disease should be taken into account. In the correlation analysis of our study, the night-time estimates were negatively correlated with NO_2_, O_3_, CO, ad PM_10_ ad positively correlated with SO_2_. Although the correlation was weak and negligible, this result is not consistent with the result of the past studies. Only a few studies have analyzed the correlation between air pollutants and noise exposure, and the duration considered in estimating the concentration level varies among the studies. A study on the correlation between traffic noise and air pollutants showed NO_2_ positively correlated with the noise [91]. One possible explanation for the inconsistency is the disparity between the timeframe applied for deriving estimates of noise level and other air pollutants; the timeframe applied for the noise levels estimates were night-time whereas other air pollutants were not designated a specific timeframe. Additionally, night-time noise levels were estimated by each individual, while air pollutant levels were estimated by each school area. Another possible explanation is that the time periods of each air pollutant emitted are different. For example, considering the primary source of NO_2_ is road transport, the NO_2_ level would be higher in day-time compared to the night-time [92,93]. On the contrary, SO_2_ is mostly emitted from stationary sources such as factories, refineries, and power plants, which may result in relatively lower diurnal variation [93]. Even then, it is an evident fact that both air pollutants and noise exert effects on health, which prompts the analysis of the health effects of co-exposure to multiple environmental pollutants. Recently, despite the challenges of analyzing the co-exposure effect on health, many researchers are trying to analyze the effect of simultaneous exposure to multiple environmental factors. Indeed, supposing that the main source of the night-time noise estimates in our study is traffic, a future study should consider the co-exposure effect of traffic emission and noise [94]. Fourth, the survey did not include questions on noise sensitivity or noise annoyance, which is an important factor in assessing noise-induced stress and health impact [53,57,95,96]. According to Kim et al., cortisol levels are more influenced by the subject’s sensitivity to noise than by the level of chronic road traffic noise [96]. Finally, this study is merely focused on investigating the association between allergic disease and night-time noise exposure, so verification of causal relationship or confirmation of possible mechanism requires further evaluation.

Even with these limitations, our study is the first to investigate the impact of noise exposure on the incidence of allergic disease in the Republic of Korea. Additionally, we have assigned individual night-time noise estimates to each subject by implementing a reliable LUR modeling method. The night-time noise standards of the residential area are 40 dB at the general residential zone and 55 dB at the roadside zone; most of the estimated night-time noise levels exceeded these standards. Further research is required to verify our findings by generating proper predictor variables and developing new LUR models with more scientific methods in a complex urban area to elicit more precise predictions. With further evaluation, the result of our study can be used as supporting evidence when determining night-time noise limits.

## 5. Conclusions

Noise pollution has become a widely distributed concern worldwide due to ongoing urbanization and industrialization. Sound sleep is crucial for restoring the body’s energy spent during the daytime. In that sense, noise perceived during one’s sleep may induce sleep disturbance and trigger stress response interrupting the body’s normal restoration process. Various health effects could arise from noise exposure, especially at night-time. The result of our study indicates that noise exposure at night-time also increases the risk of allergic disease in children, and the mechanism involved in that process warrants further study. Future studies with some improvement on limitations described earlier would be a sufficient supporting resource when revising a guideline on night-time noise limits.

## Figures and Tables

**Figure 1 ijerph-19-02748-f001:**
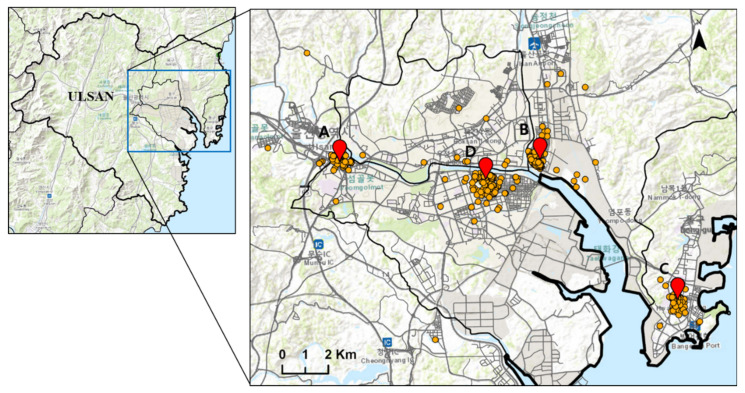
Location of 4 elementary schools: A, Gulhwa; B, Myeongchon; C, Yangi; D, Samshin.

**Figure 2 ijerph-19-02748-f002:**
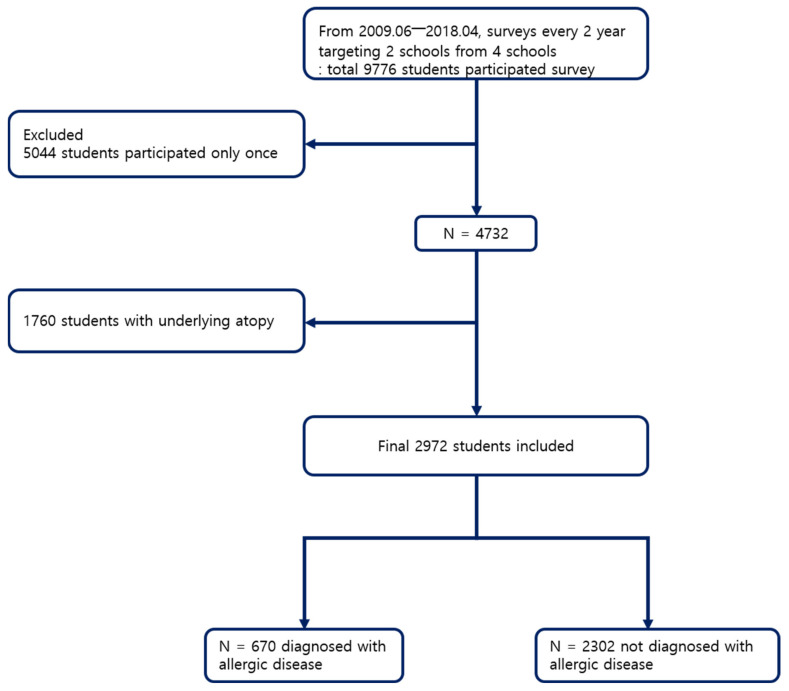
Flow chart of the subject selection process.

**Figure 3 ijerph-19-02748-f003:**
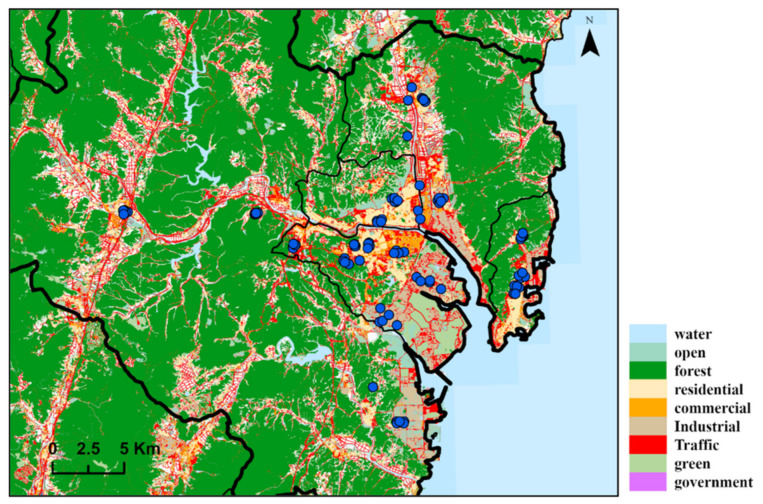
The area depicted by land use and 87 noise monitoring sites in Ulsan.

**Figure 4 ijerph-19-02748-f004:**
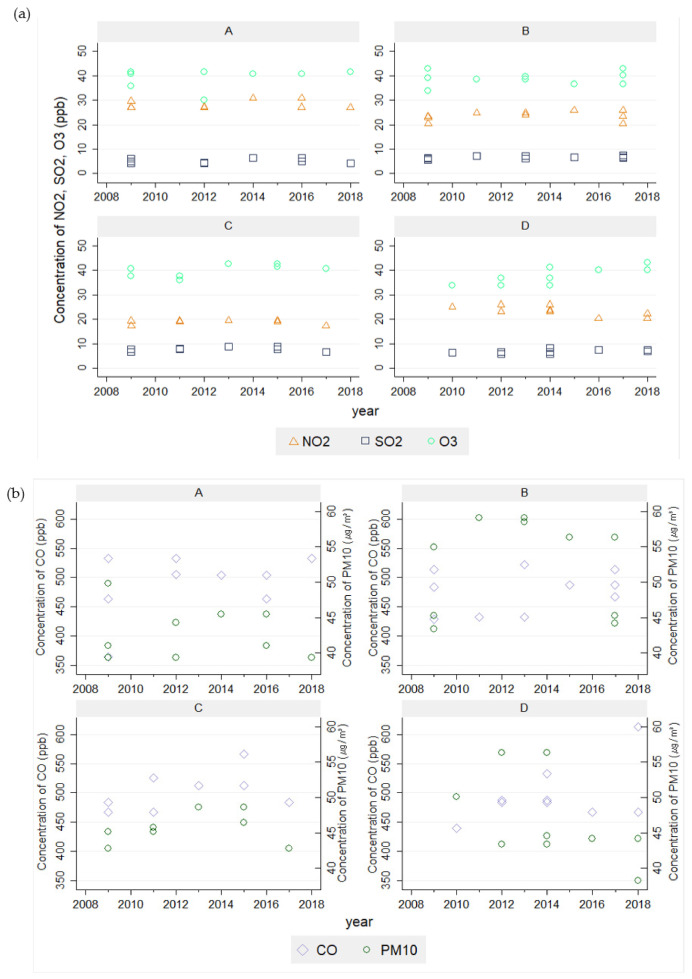
Air pollutant concentration each year by the school; (**A**) represents the coastal environment, (**B**) represents the residential area, (**C**) represents the industrial area, (**D**) represents the downtown area. (**a**) estimated concentration of NO_2_, SO_2_, O_3_ without showing a distinct pattern (**b**) estimated concentration of CO and PM10 without showing a distinct pattern.

**Figure 5 ijerph-19-02748-f005:**
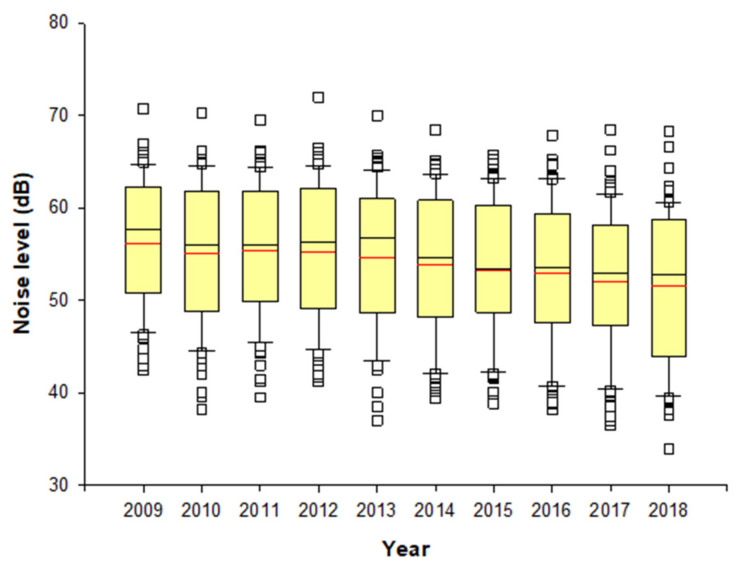
Distribution of annual (2009–2018) noise levels at each location of participants. Yellow box shows the interquartile range of estimated noise level, redline is the mean value, black line is the median and blank squares are the outliers.

**Figure 6 ijerph-19-02748-f006:**
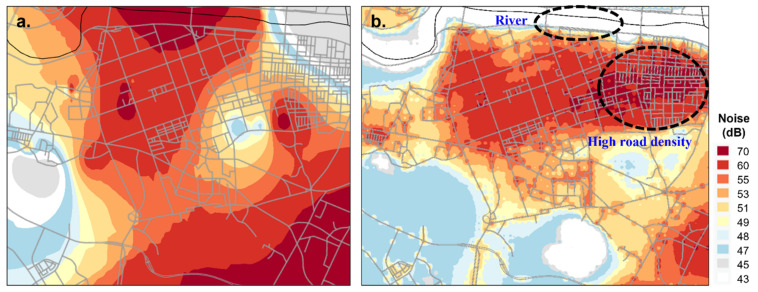
Noise mapping of part of Ulsan region (**a**) noise map based on noise level estimated from Kriging method, (**b**) noise map based on noise level estimated from LUR model.

**Table 1 ijerph-19-02748-t001:** Demographic features of subjects focused on intrinsic factors ^§^.

Variables		No Event		Event		*p*-Value
Time to event ^a,^* (days)		961.7	(356.5)	865.2	(295.7)	0.000 *
Age ^a,^* (years)		7.6	(1.20)	7.3	(1.10)	0.000 *
Sex ^b^	Male	1120	(48.7%)	348	(52.0%)	0.140
	Female	1181	(51.3%)	321	(48.0%)	
History of Allergic disease, Father ^b^	No	1777	(77.2%)	454	(67.8%)	0.000 *
	Yes	525	(22.8%)	216	(32.2%)	
History of Allergic disease, Mother ^b^	No	1680	(73.0%)	422	(63.0%)	0.000 *
	Yes	622	(27.0%)	248	(37.0%)	
History of bronchiolitis ^b^	No	2091	(90.8%)	559	(83.4%)	0.000 *
	Yes	211	(9.2%)	111	(16.6%)	
Income ^b^	<1.0	21	(0.9%)	9	(1.3%)	0.060 *
(million Korean won)	1.0~2.0	190	(8.3%)	31	(4.6%)	
	2.0~3.0	673	(29.4%)	204	(30.6%)	
	3.0~4.0	635	(27.7%)	188	(28.2%)	
	4.0~5.0	452	(19.7%)	139	(20.8%)	
	>5.0	321	(14.0%)	96	(14.4%)	
Lactation ^b^	No	507	(22.0%)	138	(20.6%)	0.460
	Yes	1794	(78.0%)	532	(79.4%)	

^§^ Independent *t*-test was performed for the continuous variables and chi-squared test for categorical variables; ^a^ mean (±standard deviation, SD); ^b^ frequency (percentage,%); * *p*-value less than 0.05 shows significance.

**Table 2 ijerph-19-02748-t002:** Demographic features of the subjects focused on environmental or extrinsic exposure variables ^§^.

Variables		No Event		Event		*p*-Value
Secondhand smoke	No	1800	(78.2%)	525	(78.4%)	0.960
	Yes	502	(21.8%)	145	(21.6%)	
Air purifier	No	1694	(73.7%)	466	(69.6%)	0.030 *
	Yes	604	(26.3%)	204	(30.4%)	
Humidifier	No	1678	(73.1%)	482	(72.0%)	0.620
	Yes	618	(26.9%)	187	(28.0%)	
Pet	No	1828	(79.5%)	535	(79.9%)	0.870
	Yes	471	(20.5%)	135	(20.1%)	
Vicinity to road	Near	365	(15.9%)	112	(16.7%)	0.910
(unit: meter)	<50	414	(18.0%)	123	(18.4%)	
	<100	608	(26.4%)	173	(25.9%)	
	<200	518	(22.5%)	149	(22.3%)	
	<300	217	(9.4%)	69	(10.3%)	
	<400	104	(4.5%)	24	(3.6%)	
	<500	75	(3.3%)	19	(2.8%)	
Amount of traffics ^†^	Low	178	(7.7%)	36	(5.4%)	0.120
	Medium	885	(38.4%)	261	(39.1%)	
	High	1239	(53.8%)	371	(55.5%)	

^§^ Chi-squared test was performed for categorical variables; * *p*-value less than 0.05 shows significance; ^†^ The amount of traffic was assessed by question “How much traffic is on the road adjacent to your house?” which subjects subjectively answered by Low, Moderate, and High.

**Table 3 ijerph-19-02748-t003:** Assessment of correlation between night-time noise estimates and air pollutants.

Variable	Noise	NO_2_	SO_2_	O_3_	CO	PM_10_
Noise	1	−0.296 **	0.231 **	−0.370 **	−0.079 **	−0.191 **
NO_2_	−0.296 **	1	−0.510 **	−0.220 **	−0.161 **	0.037 **
SO_2_	0.231 **	−0.510 **	1	0.463 **	0.186 **	0.095 **
O_3_	−0.370 **	−0.220 **	0.463 **	1	0.351 **	−0.070 **
CO	−0.079 **	−0.161 **	0.186 **	0.351 **	1	−0.445 **
PM_10_	−0.191 **	0.037 **	0.095 **	−0.070 **	−0.445 **	1

** Correlation is significant at the 0.01 level (2-tailed).

**Table 4 ijerph-19-02748-t004:** Association between night-noise difference and incidence of allergic disease.

Variable		Univariate Model					Multivariate Model ^§^				
		HR	95% CI			*p*-Value	HR	95% CI			*p*-Value
Night noise difference ^a^		1.077	1.057	–	1.099	0.000	1.710	1.424	–	2.052	0.000
Age		1.243	1.154	–	1.339	0.000	1.167	1.078	–	1.263	0.000
History of Allergic disease, Father		1.577	1.341	–	1.854	0.000	1.460	1.225	–	1.740	0.000
History of Allergic disease, Mother		1.428	1.220	–	1.670	0.000	1.364	1.152	–	1.615	0.000
O_3_ difference		0.956	0.936	–	0.977	0.000	0.942	0.919	–	0.966	0.000
CO difference		0.998	0.997	–	1.000	0.056	0.996	0.994	–	0.998	0.000
NO_2_ difference		1.047	1.002	–	1.095	0.042	1.088	1.039	–	1.139	0.000
Income ^b^	<1.0	Reference				0.042	Reference				0.020
	1.0~2.0	0.761	0.172	–	0.761	0.172	0.298	0.140	–	0.634	0.002
	2.0~3.0	1.217	0.320	–	1.217	0.320	0.518	0.265	–	1.014	0.055
	3.0~4.0	1.104	0.290	–	1.104	0.290	0.449	0.229	–	0.879	0.020
	4.0~5.0	1.051	0.273	–	1.051	0.273	0.445	0.226	–	0.879	0.020
	>5.0	1.090	0.278	–	1.090	0.278	0.462	0.231	–	0.925	0.029
Air purifier		1.236	1.048	–	1.457	0.012	1.225	1.031	–	1.456	0.021
History of bronchiolitis		1.702	1.388	–	2.087	0.000	1.538	1.243	–	1.902	0.000

^a^ The baseline is the negative difference; ^b^ unit: million Korean won. ^§^ Adjusted for age, sex, income, air purifier, parental history of allergic disease, history of bronchiolitis within 2 years of birth, lactation, exposure to second-hand smoking, NO_2_ difference, SO_2_ difference, CO difference, O_3_ difference, PM_10_ difference, ever own a pet, amount of traffic, distance to road, humidifier.

**Table 5 ijerph-19-02748-t005:** Association between night-noise difference and incidence of allergic disease.

Variables		Univariate Model					Multivariate Model ^§^				
		HR	95% CI			*p*-Value	HR	95% CI			*p*-Value
Night-noise difference rate (%) ^a^	<−5.74	Reference				0.000	Reference				0.000
	−5.74~−1.82	1.016	0.822	–	1.255	0.886	0.989	0.794	–	1.233	0.924
	−1.82~1.04	1.627	1.310	–	2.021	0.000	1.313	1.032	–	1.669	0.026
	>1.04	2.006	1.610	–	2.499	0.000	1.724	1.360	–	2.185	0.000
Age		1.243	1.154	–	1.339	0.000	1.170	1.081	–	1.267	0.000
History of allergic disease, Father		1.577	1.341	–	1.854	0.000	1.467	1.231	–	1.750	0.000
History of allergic disease, Mother		1.428	1.220	–	1.670	0.000	1.370	1.156	–	1.624	0.001
O_3_ difference rate		0.956	0.936	–	0.977	0.000	0.979	0.970	–	0.987	0.000
CO difference rate		0.998	0.997	–	1.000	0.056	0.982	0.974	–	0.989	0.000
NO_2_ difference rate		1.021	0.997	–	1.045	0.086	1.022	1.011	–	1.033	0.000
Income ^b^	<1.0	Reference				0.042	Reference				0.019
	1.0~2.0	0.761	0.172	–	0.761	0.172	0.284	0.133	–	0.604	0.001
	2.0~3.0	1.217	0.320	–	1.217	0.320	0.496	0.253	–	0.970	0.041
	3.0~4.0	1.104	0.290	–	1.104	0.290	0.440	0.224	–	0.863	0.017
	4.0~5.0	1.051	0.273	–	1.051	0.273	0.438	0.222	–	0.865	0.017
	>5.0	1.090	0.278	–	1.090	0.278	0.443	0.222	–	0.887	0.022
Air purifier		1.236	1.048	–	1.457	0.012	1.229	1.034	–	1.406	0.019
History of bronchiolitis		1.702	1.388	–	2.087	0.000	1.504	1.215	–	1.861	0.000

^a^ interquartile range of night-noise difference rate (%); ^b^ unit: million Korean won. ^§^ adjusted for age, sex, income, air purifier, parental history of allergic disease, history of bronchiolitis within 2 years of birth, lactation, exposure to second-hand smoking, NO_2_ difference, SO_2_ difference, CO difference, O_3_ difference, PM_10_ difference, ever own a pet, amount of traffic, distance to road, humidifier.

## Data Availability

Please contact author for data requests.

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
