# Peer review of "What Is the Role of Night-Time Noise Exposure in Childhood Allergic Disease?"

_ijerph, 2022, doi:10.3390/ijerph19052748_

Round 1
Reviewer 1 Report
[ "ijerph-1534696 - What’s the role of night-time noise exposure in atopic disease?". ]
This manuscript is original article about evaluation of the role of night-time noise exposure in atopic disease, and this issue has impact on the clinical fields, if below-mentioned points will be corrected and revised.
1) The authors used term "atopic disease" throughout this manuscript including the title. Although the term "atopy" has been generally used among public, this term should be described with specified definitions (e.g. "atopic diseases including allergic rhinitis, asthma, atopic eczema, etc." like as the first sentence of the Introduction of this manuscript), especially in medical papers. So the use of term "allergic diseases" instead of "atopic disease" may be more prefarable and accurate for academic papers. And the subjects of this study were all children, the authors may change the title as "What’s the role of night-time noise exposure in childhood allergic disease?"
2) In the Page 1, Line 37~42 sentences need to be added those references.
3) In the Page 1, Line 42~43 sentence need to be rewritten grammarly.
4) In the Page 2, Line 47: The "harmful inflammation" may be replaced by "harmful chronic inflammation".
5) In the Page 2, Line 47~49 sentence omitted its references.
6) In the Page 2, Line 64 sentence omitted its references.
7) In the Page 2, Line 70~72, Line 73~74, Line 78~79 sentences omitted those references.
8) In the Page 3, Line 104~107 and Page 6, Line 172~173, the questionnaire term "atopic disesease" need to be specifically defined by relevant disease entities as above mentioned comments #1.
9) In the Page 7, Line 218~219 sentence, the mean followup period "865.2 months (ranging from 2 to 3 rounds)" need to be explained. According to the Table 1, the subjects' mean age was 7.6 years old (91.2 months), so the authors should explain the exact meaning of the
above sentence.
10) On the Page 7, footnotes of the Table 1. are described in the wrong location and should be corrected. And the authors should add the units / measurment descriptions of "Age" and "Amount of traffics".
11) On the Page 9, Figure 4 need to be cleared of unnecessary lines (MS Excel remnant lines, maybe). And its legend should describe the comments on the subject-school names (eg. locations, characters, etc.) for readers. The authors may describe the school names as School A, B, C, D for relevant institutional privacy.
12) On the page 11, the Figure 7 map may contain markings of the location of the four subject-schools for reader.
13) In the Table 1 and 4, the authors may corrected the unit of Income as "million Korean Won" and add the conversion to international monetary, if possible, for international readers.
14) Considering too many Figures, the authors need to be merge several relevant figure together.
15) In the Discussion, the authors need to discuss on their results about night-time noise and common allergic diseases (allergic rhinitis, asthma) other than atopic dermatitis.
16) In the Discussion, the authors need to discuss on negative correlations between noise and air pollutants (NO2, O3, CO, PM10) and positive correlation between noise and SO2 shown in Table 3.
17) In the Conclusion, line 488 sentence, "From our result, noise exposure at night-time also increase the risk of atopic disease in children. Mechanism involved in that process warrants further study." should be rewritten properly and grammarly.
18) In the Conclusion, line 491 sentence "This section is mandatory." is should be removed.
19) English correction by native speaker is recommended.
Thank you!
Reviewer 2 Report
In my opinion, the manuscript is ready to be published after a minor revision.
Environmental noise ranks second as an environmental public health hazard. The human ear is
extremely sensitive and has virtually no rest. Even when we are asleep, our ears still perceive
sounds and the brain processes them. Living in an environment of noise has many health
consequences. Noise negatively affects not only the hearing organ but also many systems,
including: circulatory, endocrine, nervous and digestive systems. In addition to the intensity of
the sound, the negative impact of noise on the body also depends on the duration of exposure.
Both of these factors together give the so-called noise dose. Unfortunately, environmental noise
is a global problem that is difficult to eliminate. Noise can not be completely avoided, but it is
worth protecting against excess noise.
The authors of the study showed that night exposure to noise increases the risk of atopic skin
disease. These findings can be used as supporting data for defining noise limits.
In my opinion, this is a very interesting and very practical manuscript that deserves to be
published.
However, some further improvements should be made for this article to be acceptable for
publication.
1. In Table 1 the dimension is missing in the entries: Time to event , Age.
2. In the References, the authors did not take into account the required citation style:
Abbreviated Journal Name (italics) Year (bold), Volume (italics).
3. Non-uniform style of reference for table and figure numbers (unnecessary full stop
after a number ): Line 221 (Table 1.), Line 226 (Table 2.), Line 231 (Fig. 4.)
Line 274 (Table 3.), Line 294 (Table 1.), Line 299 (Table 5.)
4. In figure titles, the authors use different styles: bold or normal when numbering
figures, e.g. Figure 5, Figure 6.
5. Line 361- please check the accuracy of the citation of item [59]
Round 2
Reviewer 1 Report
The revised manuscript "ijerph-1534696-v2 - What’s the role of night-time noise exposure in childhood allergic disease?" seems to have been properly revised and corrected according to my comments.
So I think this revised manuscript can be considered to be published in this respectable journal.
Good luck !